# Exploring Carcinoid Syndrome in Neuroendocrine Tumors: Insights from a Multidisciplinary Narrative Review

**DOI:** 10.3390/cancers16223831

**Published:** 2024-11-14

**Authors:** Matteo Marasco, Elena Romano, Giulia Arrivi, Daniela Prosperi, Maria Rinzivillo, Damiano Caruso, Paolo Mercantini, Michele Rossi, Antongiulio Faggiano, Francesco Panzuto

**Affiliations:** 1Digestive Disease Unit, Sant’ Andrea University Hospital, ENETS Center of Excellence, 00189 Rome, Italy; matteo.marasco@uniroma1.it (M.M.); elena.romano@uniroma1.it (E.R.); mrinzivillo@ospedalesantandrea.it (M.R.); 2PhD School in Translational Medicine and Oncology, Department of Medical and Surgical Sciences and Translational Medicine, Faculty of Medicine and Psychology, Sapienza University of Rome, 00185 Rome, Italy; 3Oncology Unit, Sant’ Andrea University Hospital, ENETS Center of Excellence, 00189 Rome, Italy; giulia.arrivi@uniroma1.it; 4Nuclear Medicine Unit, Sant’ Andrea University Hospital, ENETS Center of Excellence, 00189 Rome, Italy; dprosperi@ospedalesantandrea.it; 5Radiology Unit, Sant’ Andrea University Hospital, ENETS Center of Excellence, 00189 Rome, Italy; damiano.caruso@uniroma1.it; 6Department of Medical and Surgical Sciences and Translational Medicine, Sapienza University of Rome, 00185 Rome, Italy; paolo.mercantini@uniroma1.it (P.M.); michele.rossi@uniroma1.it (M.R.); 7Surgery Unit, Sant’ Andrea University Hospital, ENETS Center of Excellence, 00189 Rome, Italy; 8Interventional Radiology Unit, Sant’ Andrea University Hospital, ENETS Center of Excellence, 00189 Rome, Italy; 9Endocrinology Unit, Sant’ Andrea University Hospital, ENETS Center of Excellence, Department of Clinical and Molecular Medicine, Sapienza University of Rome, 00189 Rome, Italy; antongiulio.faggiano@uniroma1.it

**Keywords:** neuroendocrine tumor, carcinoid syndrome, cancer, multidisciplinary team

## Abstract

Carcinoid syndrome in patients with neuroendocrine tumors is a challenging condition that requires accurate diagnostic and therapeutic management by a multidisciplinary team. Therefore, this review intends to detail all clinical aspects of carcinoid syndrome in order to provide clinicians with a definite overview of the effective approach to carcinoid syndrome. In this way, the review aims to improve patient care and treatment outcomes in this patient population.

## 1. Introduction

Heterogeneous biological behaviors with different clinical presentations and therapeutic approaches characterize neuroendocrine neoplasms (NENs). From a histological point of view, they can be distinguished based on the morphology of the tumor cells into well-differentiated forms, known as neuroendocrine tumors (NETs), and poorly differentiated forms, referred to as neuroendocrine carcinomas (NECs). From a clinical perspective, they can be categorized as functioning (when a specific tumor-related syndrome is present) or non-functioning (in cases where only generic, non-specific symptoms are observed).

One of the most representative clinical manifestations of functioning NETs is represented by carcinoid syndrome (CS), which is caused by tumoral secretion of multiple hormonal amines and peptides, mainly serotonin (5hydroxytryptamine, 5-HT) [1].

CS is predominantly described in about 20% of patients with well-differentiated small intestine NETs, representing 50% of all gastro-entero-pancreatic neuroendocrine tumors (GEP-NETs) [2,3]. It is often diagnosed in the fifth and sixth decades of life, with a slightly higher incidence among females than males [4]. Since the liver has the function of deactivating tumor-released active hormones, generally, CS indicates the presence of a metastatic tumor outside of the portal venous drainage; however, in a small proportion of patients, CS may develop without liver metastases [5]. Patients with CS have an overall survival of 4.7 years, compared to 7.1 years in patients without symptoms of CS, and tumor burden is described as one of the most relevant factors affecting CS-related mortality [2].

CS may occur with a vast and not-so-specific spectrum of symptomatology that is represented mainly by diarrhea and flushing because of the release of serotonin [1]. Moreover, CS in patients with NET is associated with a worse quality of life (QoL), particularly compromised by increased diarrhea or flushing [6]. As the disease progresses slowly, long-term complications may arise, such as carcinoid heart disease (CHD), fibrotic small bowel obstruction, and, importantly, malnutrition. The condition can worsen to a life-threatening complication known as a Carcinoid Crisis, which is characterized by bronchospasm, flushing, and significant hemodynamic instability (systolic blood pressure <80 or >180 mmHg, heart rate > 120 bpm) [4,7].

## 2. Focus of the Review

According to the latest scientific evidence, this review will focus on the diagnostic and therapeutic management of carcinoid syndrome in NETs. The intent is to provide clinicians with a broad and straightforward overview of the effective approach of CS, which requires a combination of biochemical, imaging, and therapeutic strategies to control both the tumor and the hormonal secretion. Through this focused approach, the review seeks to contribute valuable knowledge to enhance patient care and improve treatment outcomes in this patient population.

In this narrative review, we gathered data by conducting a comprehensive search of the MEDLINE database without imposing any date limitations. Our search criteria were conducted through specific keywords, as follows: “neuroendocrine tumors”, “carcinoid syndrome”, “diagnosis”, and “treatment”. The scope of our inclusion was limited to articles pertinent to this review’s aims and those composed in English. This research did not adhere to the systematic review protocol; instead, the articles were selected based on the authors’ subjective judgment.

## 3. Clinical Presentation

A clinical evaluation of patients with NET, in whom there is a suspicion of CS, may be challenging. However, some signs and symptoms could guide therapeutic decisions. 

CS is characterized frequently by diarrhea (60–80%) and flushing (90%), because 5-HT acts through mechanisms of vasoconstriction or vasodilation, increased gut motility, and increased secretion of water, sodium, chloride, and potassium [1].

Diarrhea is defined in terms of alterations of stool frequency, consistency, volume, or weight, and it is typically secretory, with patients complaining of at least three and up to thirty bowel movements in a day. To exclude other concomitant causes of diarrhea, personal and family history, previous surgery, and home therapy, other NET-related conditions (e.g., steatorrhea secondary to somatostatin analogs) must be investigated [4,8,9].Flushing is the most common sign of CS. The phenomenon may present as either an intermittent or persistent sensation of warmth accompanied by skin erythema, typically affecting the upper body region (head, neck, upper chest), with eventual telangiectasia. In CS, flushing usually occurs as “dry”, and is reddish brown/bright red, with short-lived episodes [5].Bronchospasm is a rare manifestation of CS (around 15% of cases). It is characterized by the development of wheezing and dyspnea; histamine and serotonin release are responsible for bronchoconstriction and local edema in the airways [4,5].

Recent reports indicate that gender-related disparities are observed in patients with CS that is associated with NET, highlighting significant variations in clinical presentation, risk factors, and outcomes between males and females. [10]. Women were found to experience more severe symptoms, including higher incidences of abdominal pain, tachycardia, and psychiatric disorders such as depression. At the same time, men showed a greater likelihood of lymph node metastases at diagnosis. Risk factor analysis indicated that men were more likely to be smokers and alcohol drinkers, which may contribute to these differences in disease manifestation. Despite these disparities in clinical presentation, the study found no significant differences in progression-free survival (PFS) or overall survival (OS) between genders, with both men and women responding similarly to treatments such as surgery and medical therapies. The data indicate that although CS occurs slightly more often in men, women might experience more severe symptomatology, highlighting the necessity for a gender-specific therapy strategy to enhance patient outcomes [10].

In the landscape of NET, women also tend to have better survival rates and treatment responses, but the disease impacts quality of life differently for both sexes. Despite known sex differences, they are primarily underestimated in clinical practice, and there is a lack of clinical trials addressing these disparities. More systematic studies on sex-related differences are needed to develop tailored treatment strategies that improve both the prognosis and quality of life in patients with CS [11].

## 4. Diagnosis

The diagnosis of CS results from a combination of appropriate clinical, biochemical, and radiological evaluations. 

When CS is suspected, the most reliable initial assessment is the measurement of 24-h urinary 5-hydroxyindoleacetic acid (5-HIAA) levels (sensitivity 73–91%, specificity 100%), as a metabolite of serotonin (>50 μmol, the cut-off to consider compatible with the diagnosis of CS) [12,13]. In selected cases, the assessment of plasma 5-HIAA can be used as a convenient alternative to the urinary 5-HIAA collection due to the similar sensitivity, the absence of influence by meals, and the advantage of a single determination in a day [5,14].

Recently, a close correlation between plasma and serum 5-HIAA has been demonstrated, with a sensitivity of 91.2% and specificity of 61.9% at a cut-off of 135 nmol/L, compared to the urinary assay [15]. A statistically significant agreement was observed between plasma, serum 5-HIAA, and the traditional urine assay in patients with nNETs (κ = 0.675, *p* < 0.001) and in healthy volunteers (κ = 0.967, *p* < 0.001), indicating that either plasma or serum can be reliably used for monitoring 5-HIAA. Based on these data, plasma measurement can be considered an alternative to urine testing. However, the latter remains more readily available in clinical practice and better standardized regarding result interpretation.

Therefore, typical clinical symptoms, such as diarrhea and flushing, and the recognition of elevated serotonin metabolite 5-HIAA in a 24-h urine test establish the diagnosis of CS [16].

Imaging studies (CT scan, MRI, and somatostatin receptors imaging, including 68Ga-DOTATATE PET, 68-DOTATOC PET, or Cu64-DOTATATE PET) have a key role in the whole assessment of the patients with NET associated with CS to determine the global tumor burden and liver metastasis, the somatostatin receptor status, and to evaluate tumor growth rate and eventual locoregional therapies [17].

Carcinoid syndrome typically occurs when NENs, primarily from the small intestine, secrete serotonin and other vasoactive substances. These substances often bypass liver metabolism, usually due to hepatic metastases, and enter the systemic circulation, causing characteristic symptoms. However, CS can also occur in the absence of liver metastases. Halperin et al.’s study [2] underscores the unexpectedly high occurrence of carcinoid syndrome (CS) across different stages of neuroendocrine tumors (NETs), with notable rates in localized (19%) and regional (37%) small bowel NETs, suggesting possible underrecognized hepatic metastases. Strosberg et al. [18] report that CS was initially present in 30% of patients, increasing to 70% during disease progression, predominantly in metastatic small bowel NETs. Additionally, colorectal NETs demonstrated high CS prevalence, potentially due to a classification that spans midgut and hindgut origins, contrasting with the low incidence in lung NETs, possibly due to bypassing hepatic clearance [2]. This figure highlights the variability in the reported rates of carcinoid syndrome in small bowel NETs, with some series reporting prevalences as high as 70% [2,17].

## 5. Treatment

Carcinoid syndrome presents a significant challenge in clinical management due to its debilitating symptoms. Currently, several therapeutic options are available for alleviating these symptoms, which can significantly impact the quality of life of affected patients. The primary treatment modality involves somatostatin analogs, such as octreotide and lanreotide, which have been shown to provide substantial symptom relief in many cases. However, for patients who experience refractory symptoms, additional treatment options exist, including telotristat ethyl, which explicitly targets serotonin production, and everolimus, an mTOR inhibitor that can help control tumor growth. Furthermore, liver-directed therapies and chemotherapy may be considered, depending on the extent of the disease and patient-specific factors. This multifaceted approach underscores the importance of effective personalized treatment plans in managing carcinoid syndrome (Figure 1).

### 5.1. Medical Therapy

First-line treatment for controlling the symptoms of CS is represented by long-acting release somatostatin analogs (SSAs), such as octreotide (30 mg/4 weeks) and lanreotide (60 mg, 90 mg, and 120 mg/4 weeks), which provide substantial symptom relief in CS. A meta-analysis summarized the treatment of CS patients with both analogs, reporting an achievement of CS symptom control in 66–70% of patients and a decrease in 5-HIAA levels in 45–46% of CS patients [19]. These agents also have an antiproliferative effect on tumor growth (directly with cycle arrest and apoptosis, indirectly by inhibition of angiogenesis and immunomodulation), prolonging PFS, as reported by phase III randomized controlled trials [20,21,22]. SSAs are safe and well-tolerated drugs, with only 15% of patients experiencing adverse events (AEs), which could be short-term AEs (usually mild/moderate), such as pain at the injection site, nausea (76%), constipation (85%), diarrhea (16–78%), abdominal cramps (50%), and hyperglycemia, and long-term AEs, including cholelithiasis (5–60%) and exocrine pancreatic insufficiency (20–24%) [9].

However, in about 30–76% of cases, patients with NET associated with CS may experience refractory CS (RCS) (recurrence/persistence of CS symptoms and increase/persistence of high urinary 5-HIAA levels despite the use of maximum label doses of SSA). This condition may be a result of the progression of symptoms, which is more often caused by tumor growth than by the development of tachyphylaxis to the administered doses of SSA [5,16]. Nevertheless, several other medical therapeutic strategies can be used to aid clinicians. 

The first therapeutic step in patients with RCS is to evaluate the possibility of SSA dose escalation, decreasing the interval time of SSA administration. In this regard, several studies have reported favorable clinical, biochemical, and tumor response results. Strosberg et al. reported that on 239 patients receiving above-standard doses of SSA due to CS (62%) or tumor progression (28%), the most common dose changes were an escalation to 40 mg every four weeks (71%), and to 60 mg every four weeks (18%). Of 90 patients with flushing before the first dose escalation, 73 (81%) were reported to have experienced an improvement or resolution of their symptoms following the dose escalation. Of 107 patients who complained of diarrhea before the first dose escalation, 85 (79%) were reported to have experienced an improvement or resolution after the first dose escalation [23]. Indeed, a small phase II study showed that increasing the frequency of administration of SSA to every 21 days led to complete and partial control of clinical symptoms in 40% and 60% of cases, respectively [24]. ENETS guidelines also suggested that administration of the long-acting SSA should be combined with (100–500 μg every 6–8 h), for up to 2 weeks, or as a rescue therapy when CS is not controlled [5].

Analyzing the data from the available studies is not straightforward due to various influencing factors. One study showed variability in the effectiveness of octreotide LAR for managing CS symptoms, with efficacy rates ranging from 25–70% [16]. This variation is likely attributed to differences in the study designs and inclusion criteria, including symptom severity, tumor burden, and SSA dosing schedules. Additionally, when reporting adverse AEs, it is crucial to consider potential confounding factors, such as differing toxicity assessment methods. Symptoms of CS, like abdominal pain and diarrhea, can be challenging to differentiate from those caused by SSA.

When managing a patient with refractory carcinoid syndrome, it is essential to determine whether the disease is progressive. In cases of radiologically stable disease with uncontrolled symptoms, increasing the dose of somatostatin analogs (SSAs) or adding telotristat may be beneficial. If the disease is predominantly liver-based, liver-directed therapies should be considered. Conversely, if the uncontrolled syndrome is associated with radiologically progressive disease, systemic treatments such as peptide receptor radionuclide therapy (PRRT) [25] or everolimus are indicated.

Pasireotide, a somatostatin analog with a high affinity for certain somatostatin receptors, showed effectiveness in reducing diarrhea and flushing in RCS patients, but it is not approved for treating carcinoid syndrome due to its trial discontinuation and issues like high rates of hyperglycemia [26,27].

Interferon alpha (IFN-α) is an older drug that may be considered in patients with RCS due to its cytotoxic, anti-angiogenic, and immunomodulatory effects. However, its safety profile, characterized by flu-like symptoms, chronic fatigue, and liver and bone marrow toxicity, has limited its use [28]. Although an early study reported improvement in 50% of patients treated with IFN after SSAs failed to control CS, subsequent prospective randomized trials did not confirm any additional benefit of IFN over SSA monotherapy in controlling tumor growth [15].

Telotristat ethyl is an oral inhibitor of tryptophan hydroxylase, the limiting enzyme of serotonin synthesis. The recommended dose is 250 mg three times a day [5]. The most recent studies that evaluated telotristat for CS control refractory to somatostatin analogs are two phase III studies (TELESTAR and TELECAST), using 250 mg and 500 mg three times a day. These trials described similar results regarding a decrease in bowel movement frequency, ranging from 26–43%, and a significant reduction of the urinary 5-HIAA value [29,30]. As the results of the TELEPATH study described, the main AEs registered due to telotristat administration included liver-related AEs, depression, and gastrointestinal AEs (e.g., nausea and abdominal pain). The occurrence of AEs was not related to dosage or duration of therapy. Most AEs were mild to moderate in severity, and no deaths were related to telotristat ethyl [31].

Everolimus is an mTOR inhibitor, which blocks the mammalian target of the rapamycin (mTOR) pathway, which plays a key role in regulating cell growth, proliferation, metabolism, and survival. 

The main study was the RADIANT-2, a double-blind, placebo-controlled phase III trial of 429 patients with advanced NET and a history of symptomatic carcinoid syndrome investigating the efficacy of everolimus, 10 mg daily, in combination with long-acting octreotide, consisting of 30 mg every 28 days. Although the primary endpoint was progression-free survival, a significant reduction in the urinary 5-HIAA in the everolimus arm was also reported compared to the other arm (61% vs. 36%) [32]. The control of symptoms associated with carcinoid syndrome was not assessed, but they are properly derived from case reports and case series [33]. The most frequent AEs registered from RADIANT-2 were stomatitis, rash, and fatigue.

Peptide receptor radionuclide therapy (PRRT) with ^177^Lu-DOTATATE represents an efficient and safe systemic option among the wide possibilities to treat NET patients with positive somatostatin receptor imaging, and to improve symptoms in progressive disease.

The NETTER-1 phase III trial randomized 229 patients with advanced well-differentiated G1- or G2-progressing midgut NET in two arms (Lutetium^177^ Dotatate plus octreotide LAR 30 mg vs. octreotide LAR 60 mg) [34]. Regarding CS diarrhea (present almost equally in both treatment arms), it improved equivalently in 48% and 43% of the patients in the ^177^Lu-DOTATATE + octreotide LAR 30 mg arm vs. the octreotide LAR 60 mg arm alone, respectively; however, considering diarrhea, quality of life after PRRT is significantly better than in the control arm in terms of time for deterioration. Unfortunately, there was no difference in the control of other symptoms, including flushing [35]. The study by Bongiovanni et al. showed significant improvements in the symptoms of diarrhea and flushing among patients with functioning NENs treated with ^177^Lu-DOTATATE. Of the 68 patients, 62 (91.1%) reported diarrhea and flushing as primary symptoms. After treatment, 88.1% experienced a syndromic response, substantially reducing these symptoms. Specifically, bowel movements were reduced by 30% or more over 12 weeks, while flushing episodes significantly decreased. These improvements in diarrhea and flushing contributed to an enhanced quality of life for the patients receiving ^177^Lu-DOTATATE [36].

Regarding the RCS, a recent study included 22 patients with a metastatic midgut NET, elevated urinary 5-HIAA acid excretion, and flushing and/or diarrhea despite treatment with SSA and without documented disease progression. ^177^Lu-DOTATATE was administered, with a primary aim to reduce symptoms. After PRRT, bowel movement frequency decreased more than 30% in 47% of patients, and flushing decreased more than 50% of the daily flushing in 67% of patients. A significant decrease in urinary 5-HIAA was reported in 56% of patients [37].

Concerns have been raised regarding the potential worsening of symptoms that may occur in patients with carcinoid syndrome undergoing PRRT. The study by Rico et al. highlights that although carcinoid crises, including severe diarrhea and flushing, are rare during PRRT, high-risk patients—especially those with a history of Carcinoid Crisis, large tumor burdens, or liver metastases—are vulnerable. Prophylactic use of octreotide and corticosteroids, in most cases, reduced the severity of symptoms. Despite these measures, some patients still experienced symptom flares, underscoring the need for tailored pre- and post-PRRT management protocols. The findings suggest that identifying high-risk patients early and implementing standardized prophylactic interventions can significantly improve outcomes and reduce the incidence of severe carcinoid crises [38].

Chemotherapy is rarely applied in clinical practice because of its limited role in CS symptom control. Most data available are old and scarce, and are not descriptive of CS-specific outcomes, but, unfortunately, they are restricted to the biochemical response [5]. The reported 5-HIAA response rates after chemotherapy regimens, which include combinations of streptozotocin, cyclophosphamide, platinum derivatives, or 5-fluorouracil, are 31% on average across 111 patients (ranging from 0–71%). Clinical response data are limited to cisplatin (0%), cyclophosphamide combined with methotrexate (6.7%), and lomustine with 5-fluorouracil [19].

### 5.2. Locoregional Treatment: Liver-Directed Therapies

Locoregional therapies play a significant role in the management of CS in patients with NET, especially for controlling larger or multifocal lesions in oligometastatic patients, or for hormonal symptom relief, as international guidelines suggested [3,39]. Since most of all NET patients with CS present liver metastasis, liver-directed therapies are particularly effective in controlling this syndrome. Multiple liver-directed interventions have been assessed, including radiofrequency ablation (RFA), selective internal radiotherapy (SIRT), and hepatic arterial embolization (HAE) or chemotherapy (TACE). The size, number, and distribution of liver metastases are important factors affecting the survival and choice of treatment strategies, which are often considered for CS patients with inoperable liver metastases [40].

Prospective clinical studies evaluating the symptomatic benefits of these locoregional techniques in patients with carcinoid syndrome are scarce.

The analysis of 479 patients from 25 studies showed that liver-directed interventions result in an 82% overall symptomatic response rate and a 61% reduction in serotonin-related biochemical markers, particularly 5-HIAA. This is significant, since serotonin plays a key role in causing symptoms like diarrhea and flushing in carcinoid syndrome. The interventions primarily include methods that selectively reduce blood supply to liver tumors, such as embolization techniques, which block the blood flow and reduce the tumor’s ability to secrete hormones [19].

Embolization methods, including bland embolization, chemoembolization (which combines embolization with direct chemotherapy, directly delivered to liver tumors), and radioembolization (using radioactive particles), were shown to provide high efficacy in controlling symptoms. These treatments were effective even in patients who had previously received SSAs, a common first-line therapy for managing hormone secretion. Despite the effectiveness, many studies were retrospective and lacked controls, raising concerns about potential bias.

Liver-directed therapies offer a crucial option for patients whose symptoms persist despite medical management. However, there is a clear need for prospective, high-quality trials to understand the long-term outcomes better and optimize treatment protocols [19].

One significant complication is post-embolization syndrome, which occurs in approximately 40% of patients undergoing TAE or TACE [41]. This syndrome typically presents with symptoms such as right-sided abdominal pain, nausea, vomiting, fever, leucocytosis, and transient liver function abnormalities. While these symptoms are usually temporary, they can cause considerable discomfort.

More serious complications are less common but noteworthy. These include liver abscesses, gallbladder necrosis, intestinal ischemia, and liver insufficiency. A high incidence of biliary injury has been reported in phase II studies of the drug-eluting bead TACE, with complications such as biloma and hepatic abscess being observed. Certain conditions, such as portal vein thrombosis, biliary stasis, and poor liver function, are considered contraindications for TAE and TACE. 

Selective embolization is always advisable to reduce ischemic complications.

Radioembolization, also known as SIRT, has gained traction over the past decade due to its promising results in tumor reduction and manageable toxicity profiles. The procedure involves the intra-arterial delivery of radioactive beads to deliver high doses of radiation directly to liver tumors. Recent studies indicate that radioembolization yields objective response rates ranging from 36% to 54%, with disease control rates between 69% and 94% [42]. Factors such as tumor grading and extrahepatic disease can influence the treatment’s effectiveness. Data from various studies indicate that the incidence of serious complications is relatively low. Liver damage resulting from radioembolization has been reported at varying rates by different studies, with some indicating a very low risk (0–2%) [41], while others report higher percentages, up to an 8% likelihood of severe hepatic toxicity [43]. High liver tumor volumes, such as over 50%, may predispose patients to chronic radiation hepatitis due to widespread radiation dispersal in the liver, a concern especially for patients with a long life expectancy. While SIRT can be advantageous over TAE/TACE in cases of localized vascular tumors with high radioactive bead uptake, its long-term risks, particularly in patients with bilobar metastases, are notably concerning [44]. Furthermore, the overall complication rates associated with this procedure are generally reported to be manageable, with most patients experiencing only mild to moderate side effects. Serious complications, such as radiation pneumonitis or gastrointestinal ulcers due to extrahepatic deposition of radioactivity, are rare, but can occur. The risk of lung shunting, where radioactive particles inadvertently travel to the lungs, is a critical concern; however, thorough pre-treatment imaging and planning can mitigate this risk effectively.

### 5.3. Surgery

Debulking surgery for liver disease should be considered a recommendable palliative option in patients with symptoms related to carcinoid syndrome refractory to medical therapy, or in whom there is evidence of clinical or radiologic progression disease, as guidelines suggested [3]. When liver metastases are potentially resectable, surgery can offer long-term disease-free survival and relief from CS symptoms. However, patients with CS often have extensive liver involvement, making complete surgical resection rarely feasible [28].

For a very select group of young patients (less than 1%) with liver-only metastases stable disease, and a resected primary tumor, liver transplantation could be considered, as studies report a 5-year recurrence-free survival rate of 20–30% [45].

A recent study evaluated the effectiveness of surgical cytoreduction in patients with small intestinal NETs that have metastasized to the liver and peritoneum [46]. Conducted over a 20-year period, the study analyzed 261 patients who underwent cytoreductive surgery, comparing outcomes between those with isolated liver metastases and those with both liver and peritoneal metastases. The findings reveal that complete cytoreduction was achieved in 78% of patients with isolated liver metastases compared to 56% of those with both types of metastases. Despite these differences, median overall survival after complete cytoreduction was similar for both groups, approximately 11.5 years for isolated liver metastases and 11.2 years for those with both liver and peritoneal metastases. Notably, the study found that 97% of patients experienced relief from carcinoid syndrome symptoms post-surgery. This data highlights that aggressive surgical approaches can lead to favorable outcomes even in patients with peritoneal metastases, suggesting that the presence of such metastases should not automatically exclude patients from surgical intervention [46]. 

The study by Chan et al. examined outcomes of cytoreductive surgery for 55 patients with low-grade NETs and extrahepatic metastases. The procedure resulted in significant hormonal and symptomatic control, with 70% of patients achieving a hormonal response and 75% experiencing symptomatic improvement. Post-operative morbidity and mortality were low, at 18% and 3.6%, respectively. Long-term outcomes were favorable, with a 5-year progression-free survival of 51% and a 5-year overall survival of 77%. The findings suggest that cytoreductive surgery can be safely performed in patients with extrahepatic NET metastases, providing effective symptom control and favorable long-term outcomes [47].

## 6. Complications

Complications of CS (Table 1) arise primarily through direct pathways from the prolonged and excessive release of hormonal amines and peptides, mostly serotonin, into the circulation, affecting multiple organ systems, and also through indirect pathways due to surgical resection or drug administration.

### 6.1. Carcinoid Crisis

The most serious complication of CS is Carcinoid Crisis (CC), a potentially life-threatening condition in which the sudden release of high levels of vasoactive hormones can result in hemodynamic instability because of distributive shock [5].

The symptoms can be dramatic, with the rapid onset of abrupt flushing, severe shifts in blood pressure, profuse diarrhea, and distressing bronchospasm with wheezing.

The exact incidence of CC is unknown, mainly because a standard definition has not been established, so the reported rates vary widely between groups [48]. The diagnosis is predominantly clinical, depending on an unexpected onset of the aforementioned symptoms in individuals with confirmed or suspected NET.

Although the pathophysiology of CC is hypothesized to be attributable to a sudden, massive release of vasoactive hormones, a prospective study conducted by Condron et al. did not find a statistically significant increase in serotonin, histamine, kallikrein, or bradykinin during the crises, suggesting that CC could be an entirely separate pathophysiologic entity from CS, rather than the extreme end of a spectrum of CS [49].

Common triggers of CC include general anesthesia, as well as procedures that result in tumor manipulation, like surgery, PRRT, tumor biopsy, or liver-directed therapy [50].

A systematic review and meta-analysis by Xu A. et al., conducted on surgical patients with midgut NETs and/or neuroendocrine liver metastasis, found that CC is relatively common, occurring in 1 in 5 patients (incidence = 19%) in the pooled data [51]. In that review, the risk of CC increased in patients with liver metastases and decreased in males. Interestingly, other characteristics traditionally considered risk factors, including CS and carcinoid heart disease, were not significantly associated with an increased risk of CC. 

Given the severity and difficulty of managing this condition, which can require an intensive care setting, preventing CC, rather than treating it once it is established, is important.

International guidelines suggest administering SSA in patients with functional NETs as a perioperative preparation to prevent CC [5,52]. Specifically, the latest ENETS guidelines recommend prophylactic administration of octreotide with a precise dose scheme, as follows: 100–500 μg subcutaneously every 6–8 h or infused at a starting dose of 50 μg/h, increased to 100–200 μg/h if necessary, 12 h pre-operatively and before anesthesia, continuously throughout the procedure, and post-operatively until the patient is clinically stable [5]. However, CC is not entirely preventable, so prompt recognition is crucial, since it can result in serious post-operative complications and, finally, death. Once established, an aggressive treatment, including intravenous fluids, corticosteroids, and vasopressors, is needed [5]. Despite guidelines strongly recommending the use of octreotide for preventing carcinoid crises, some of the literature reports suggest an opposing view. The study by Wonn et al. [53] investigates the effectiveness of perioperative octreotide in preventing carcinoid crises during surgeries for NEN. It prospectively analyzed 171 patients undergoing 195 operations without perioperative octreotide from 2017–2020, and found that omitting octreotide did not lead to increased rates or durations of crises compared to earlier studies that used the drug. The study concludes that perioperative octreotide may be discontinued due to its ineffectiveness, underscoring the need for an effective alternative to manage and mitigate the risks associated with carcinoid crises.

This necessitates further research into the mechanisms inducing carcinoid crises and continued data collection in a prospective, multicentric setting to more precisely define the optimal management strategy for patients with carcinoid syndrome undergoing surgery.

### 6.2. Mesenteric Fibrosis

Mesenteric Fibrosis (MF) is a significant and common complication in patients with CS, occurring because of prolonged release of serotonin and other bioactive substances.

Serotonin has classically been considered the main mediator of MF, since it exerts both mitogenic and fibrogenic effects in fibroblasts, smooth muscle cells, and endothelial cells.

Apart from serotonin, many other mediators have been studied as potentially responsible for MS, such as Transforming Growth Factor-α and -β (TGF-α, TGF-β), Connective Tissue Growth Factor (CTGF), Platelet-Derived Growth Factor (PDGF), Insulin-Like Growth Factor 1 (IGF-1), and Epidermal Growth Factor (EGF) [54]. At least some signs of MF occur in approximately 50% of CS patients, resulting from a fibrotic and desmoplastic reaction around metastatic mesenteric lymph nodes [55]. MF causes contraction and tightening of the adjacent bowel loop, potentially leading to acute complications, such as intestinal obstruction, intussusception, and/or intestinal ischemia because of an impaired blood supply to the intestines. Nevertheless, MF can cause chronic conditions such as postprandial abdominal pain, thus influencing patients’ food intake and resulting in malnutrition and weight loss in the long term. Another rare complication of MF is diffuse retroperitoneal fibrosis, which happens when tumor secretory products drain directly into the systemic circulation through retroperitoneal lymphatic spread. This may lead to urinary system obstruction and renal failure late in the disease clinical course [56]. 

A retrospective cohort study found that advanced MF associated with intestinal ischemia is a poor prognostic factor for OS [57].

MF can be radiologically identified by calcified mesenteric mass, fibrotic radiating strands showing a stellate pattern, and adjacent bowel-wall thickening [17]. A retrospective study by Pantograg-Brown L. et al. proposed a classification for the radiological severity of MS based on the number and thickness of radiating strands on CT images; however, this classification showed no correlation with any kind of clinical aspects, such as symptoms or prognosis [58].

The management of MF is mainly surgical and aimed at providing symptomatic relief. However, locoregional surgical resection has to be carefully considered, since it is technically challenging and carries risks due to vascular involvement and the extensive nature of fibrosis.

### 6.3. Malnutrition

Malnutrition is a common condition in oncological diseases and is associated with poorer responses to treatment, increased rates of complications, and a decreased quality of life. Maasberg et al. demonstrated malnutrition’s existence and potential prevalence in people with a GEP NET, assessing that malnutrition was diagnosed in 25% of patients. Patients with Grade 3 disease had a significantly higher prevalence of malnutrition than patients with Grade 1 or 2 disease (57.9 and 22.1%, respectively; *p* = 0.002). Lastly, it is reported that malnourished patients had a significantly shorter overall survival (19.94 vs. 31.17 months, *p* < 0.001) and a significantly longer length of stay than well-nourished patients (8.8 vs. 4.0 days, respectively; *p* < 0.001) [59]. 

Patients with GEP NETs, particularly those associated with CS, may have a nutrition and metabolic asset that is altered due to the following multifactorial causes: excessive hormones and peptide release, signs and symptoms related to the eventual progression of the disease, vitamin deficiency, and treatments administered. However, there is limited information and clinical awareness about nutrition issues in these patients.

Chronic diarrhea is one of the burdensome signs that characterized CS clinical presentation and affected NET patients’ quality of life, especially in those who experienced ≥4 bowel movements per day [60]. It could be mainly caused by tumoral excessive serotonin release and surgical intestinal resection, which could be responsible for reduced absorption of bile acid and cobalamin (B12 vitamin), and reduced bowel motility. Chronic diarrhea can lead to loss of appetite, electrolyte imbalances, dehydration, and short gut syndrome or bacterial overgrowth in a case of surgical intestinal resection [61]. Antidiarrheal agents, such as loperamide, may be useful [5].

As mentioned above, the main therapeutic approach in CS consists of SSA administration. Considering the inhibitory effect of SSAs on pancreatic exocrine function, long-term SSA administration could lead to reduced secretion of digestive enzymes, causing symptoms like steatorrhea, weight loss, and malabsorption of lipids and liposoluble vitamins (A, D, E, and K). This clinical and biochemical presentation describes a condition of exocrine pancreatic insufficiency (EPI) with a prevalence rate of approximately 20–24% in patients receiving long-term SSA treatment [9,62]. The dosage of fecal elastase (FE-1 < 200 µg/g, diagnostic cut-off) represents one of the indirect tests, and it is non-invasive, with high sensitivity and specificity for the detection of EPI [63]. Therapeutic management of EPI includes treatment with pancreatic enzyme replacement therapy (PERT) at the minimum dosage of at least 40,000 USP units of lipase during each meal. Indicators of successful treatment with PERT include reduced steatorrhea and related gastrointestinal symptoms, weight gain, increased muscle mass and function, as well as improved levels of fat-soluble vitamins [64]. Naturally, primitive pancreatic resection is another cause that can result in EPI. It depends on the type and the extension of resection; indeed, the post-operative EPI incidence is 56–98% in pancreaticoduodenectomy and 12–80% following distal and central pancreatectomy [65].

Moreover, another characteristic risk factor in NET patients with CS may be supported by malnutrition due to elevated levels of serotonin or its precursor (5-hydroxytryptophan). Under normal conditions, only a small portion of tryptophan metabolism is dedicated to serotonin production, with the majority following a pathway that partially converts it to niacin (B3 vitamin). However, the increased shift of tryptophan toward serotonin synthesis in CS patients may lead to varying degrees of niacin deficiency [66]. The prevalence of biochemical or “sub-clinical niacin deficiency” may be as high as 30–45% [61]. The most reliable and sensitive test to define niacin status is urinary excretion of N1-methyl nicotinamide and its derivative N1-methyl-2-pyridone-5-carboxyamide, as Bouma et al. utilized in a recent study to assess the niacin status before and after supplementation in patients with serotonin-producing NET [66,67]. Furthermore, we know that eventual niacin deficiency has a relevant clinical role in the management of CS patients because of the risk of developing pellagra. The clinical manifestations of pellagra are scaly skin, glossitis, stomatitis, and confusion. Violent behavior and language deficits are also associated with niacin deficiency. In isolated cases, this condition could lead to death [4].

For this reason, searching for conditions of sub-clinical niacin deficiency and contributing to niacin supplementation is a crucial clinical aim. Current evidence suggests that it would be appropriate to recommend proactive niacin supplementation of at least 40 mg per day for patients with CS and at least 100 mg per day (200–250 mg) for those with confirmed niacin deficiency [5,61].

Furthermore, some studies focused on the role of sarcopenia in NET patients as another factor in the malnutrition overview. A recent retrospective study included 30 patients with advanced, well-differentiated gastrointestinal G1 and G2 NET, and, through CT scan evaluation with a calculation of the skeletal muscle index, it was found that 66% of patients both at the time of diagnosis and follow-up presented a sarcopenia status with statistical significance in the group of patients with carcinoid syndrome (*p* = 0.0178), EPI (*p* = 0.0018), and weight loss (*p* = 0.0001) [68].

Therefore, in light of the above, it is crucial to increase clinical awareness about the nutritional status of NET patients, particularly associated with CS. Physicians should evaluate clinical signs (e.g., bowel movement frequency, weight, skin evaluation), nutritional and metabolic assets (liposoluble vitamins, niacin, B12 vitamin, anemia, blood glucose, cholesterol, albumin), and exocrine pancreatic function (fecal elastase), as well as suggesting nutritional counseling to the patient.

### 6.4. Carcinoid Heart Disease

Carcinoid heart disease (CHD) is a complex cardiac complication that mainly involves the right-sided heart valves, eventually leading to right heart failure (RHF).

CHD is present in approximately 20–50% of CS patients and significantly affects prognosis, with a reduced overall survival rate of 31% at three years in patients with CHD, compared to 69% in patients without CHD [5,69].

The right heart is mainly affected due to its direct exposure to the bloodstream from the systemic venous circulation, and to the hormones and bioactive substances produced by the neuroendocrine tumor.

CHD is characterized by a plaque-like deposition of fibrous tissue on valvular cusps leaflets, papillary muscles, chordae, ventricular walls, and occasionally on the intima of pulmonary arteries [70]. Fibrosis causes valve thickening, restricting valve mobility and resulting in regurgitation, stenosis, or both. The tricuspid valve is typically the most affected, with tricuspid regurgitation being a common consequence, while the pulmonary valve may also exhibit regurgitation or stenosis.

Left heart valves are usually spared because the vasoactive substances responsible for CHD are enzymatically inactivated in the lung vasculature, preventing transport to the left heart. However, the left-sided disease can occur in <10% of patients in the presence of a patent foramen ovale (which causes a right-to-left atrial shunt), in the presence of a functioning lung NET, or in the presence of high serotonin levels due to poorly controlled carcinoid syndrome overwhelming the hepatic and pulmonary degradative capacity [71].

All patients with CS should be questioned and examined for symptoms and physical signs of CHD, which include fatigue, dyspnoea, peripheral edema, and/or ascites.

A recent systematic review and meta-analysis, including 36 articles, showed that N-terminal pro-brain natriuretic peptide (NTproBNP) and 5-HIAA levels were higher in patients with CHD than those without CHD [72]. Therefore, ENETS guidelines recommend the evaluation of NT-proBNP as a valid screening tool for identifying patients with a high suspicion of CHD in the context of CS at baseline and follow-up [5].

Moreover, since transthoracic echocardiography (TTE) is the key investigation for CHD diagnosis and monitoring, the same guidelines recommend performing a TTE every 6–12 months in any patient with CS and/or a high 5-HIAA level with CHD symptoms and/or an NT-proBNP > 260 pg/mL. TTE should be ideally undertaken by an experienced professional in CHD.

Instead, NANETS advises that patients with substantial increases in serotonin or 5-HIAA levels (for example, more than five times the upper limit of normal) should receive an annual echocardiography, at the minimum. Screening may also be considered for patients with less pronounced elevations in serotonin levels [39].

Factors associated with the development and progression of CHD are challenging to identify, as the main publications have analyzed them with varying definitions. Several flushing episodes per day, a 5-HIAA urinary level, and a high liver burden disease have been postulated as possible factors [73].

No treatment has been reported to induce regression of the fibrous-plaque deposits, and CHD-related valve disease is currently considered irreversible. Treatment of CHD is complex and relies on the control of CS and surgical valve replacement in cases of severe CHD. A multidisciplinary approach between NET oncologists, cardiologists, cardiac surgeons, and anesthesiologists is needed, since the timing of surgery is the most crucial point of CHD management, and has to be decided on a case-by-case basis. Surgical intervention should be considered early, before the disease progresses to a point at which surgery would be too high-risk, and with at least 12 months of anticipated post-operative survival from their NET disease [5].

## 7. Conclusions

At the heart of effective management of CS lies the indispensable role of a multidisciplinary team [74]. This collaborative approach is not merely beneficial, but essential in addressing the multifaceted nature of carcinoid syndrome. The team should ideally include gastroenterologists, endocrinologists, radiologists, oncologists, nuclear medicine physicians, surgeons, cardiologists, and nutritionists. Each specialist brings unique expertise, allowing for a holistic approach to patient care. This interdisciplinary collaboration is vital in addressing all pathological aspects of the condition, formulating the most appropriate therapeutic strategy tailored to each patient, and implementing early interventions to improve their overall quality of life.

The synergy created by this multidisciplinary approach ensures that patients receive comprehensive care that addresses the primary symptoms, potential complications, and comorbidities. By leveraging various specialists’ collective knowledge and skills, healthcare providers can offer more nuanced, personalized treatment plans that significantly enhance clinical outcomes and the overall well-being of CS patients.

## Figures and Tables

**Figure 1 cancers-16-03831-f001:**
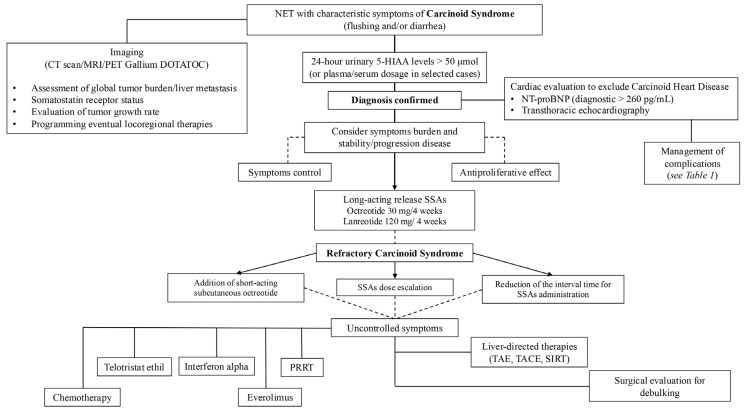
Diagnostic and therapeutic management of carcinoid syndrome.

**Table 1 cancers-16-03831-t001:** Complications management in patients with carcinoid syndrome.

Complication	Causes	Clinical Presentation	Treatment
Carcinoid Crisis (19%)	General anesthesia Surgery PRRT Tumor biopsy Liver-directed therapy	Abrupt flushing Haemodynamic instability Profuse diarrhea Bronchospasm and wheezing	Octreotide i.v. (bolus and continuous infusion), fluids resuscitation, corticosteroids, vasopressors
			Prophylactic administration of octreotide: 100–500 μg s.c every 6–8 h or i.v. at a starting dose of 50 μg/h, increased to 100–200 μg/h if necessary, 12 h pre-operatively and before anesthesia, continuously during and post-procedure.
Mesenteric Fibrosis (~50%)	Prolonged release of serotonin TGF-α, TGF-β, PDGF, IGF-1	Abdominal pain Intestinal obstruction Intussusception Intestinal ischemia	Surgery
Malnutrition (25%)	Chronic Diarrhea (Prolonged serotonin release or post-intestinal surgical resection)	Loss of appetite, electrolyte imbalances, short gut syndrome, bacterial overgrowth, cobalamin deficit	Loperamide + hydration
	EPI (SSAs-related or post-pancreatic surgical resection)	Steatorrhea, liposoluble vitamin deficit, weight loss	PERT (minimum 40,000 USP)
	Niacin deficiency	Pellagra (scaly skin, glossitis, stomatitis, and confusion)	Nicotinamide 200–250 mg
	Sarcopenia	Loss of muscle mass and strength and decline in physical functioning	Biochemical and radiological monitoring
Carcinoid Heart Disease (20–50%)	Hormones and bioactive substances from venous blood circulation	Valve thickening, restricted valve, mobility regurgitation, stenosis (tricuspid valve)	TTE and NTpro-BNP control every 6–12 months
			Surgical valve replacement in severe cases

PRRT: peptide receptor radionuclide therapy; EPI: exocrine pancreatic insufficiency, PERT: pancreatic enzyme replacement therapy; TTE: transthoracic echocardiography.

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
