# Peer review of "Exploring Carcinoid Syndrome in Neuroendocrine Tumors: Insights from a Multidisciplinary Narrative Review"

_cancers, 2024, doi:10.3390/cancers16223831_

Round 1
Reviewer 1 Report
Comments and Suggestions for Authors
This is a well-written review of carcinoid syndrome. All aspects are described appropriately and in order, and there are no problems that require any particular changes. It is also possible to accept papers in the current format.
Author Response
This is a well-written review of carcinoid syndrome. All aspects are described appropriately and in order, and there are no problems that require any particular changes. It is also possible to accept papers in the current format.
Reply: we thank the Reviewer for the comment.
Reviewer 2 Report
Comments and Suggestions for Authors
Reasonably well-written review of carcinoid syndrome, although could use some attention to language and grammar.
Comments:
1. Lines 157-165 and elsewhere: The frequency of carcinoid syndrome among patients with small bowel NETs is not well known. To start with, the diagnosis of carcinoid syndrome is often uncertain: e.g. a patient with diarrhea (which could be related to multiple factors e.g. bile salt malabsorption) and mildly elevated 5-HIAA. However, it seems to me that the analysis by Halperin et al, while often quoted, is a particularly poor method for ascertaining the frequency of carcinoid syndrome. This study was based on Medicare billing which is often random (e.g. a doctor may see a NET patient and automatically code for carcinoid syndrome, or see a patient with elevated CgA without a NET diagnosis and code for carcinoid syndrome). I would either delete this reference or point out its limitations. There are other institutional studies evaluating carcinoid syndrome frequency. For example: J Clin Oncol. 2013 Feb 1;31(4):420-5. doi: 10.1200/JCO.2012.44.5924, reported that approximately 70% of small bowel NETs (mostly metastatic) developed carcinoid syndrome over the course of their disease.
2. Line 150: Ga68 dotatoc is not the only SSTR PET imaging modality. There is also Ga68 dotatate and Cu64 dotatate
3. Line 154 please rewrite this sentence: it is highly convoluted
4. Line 198: I wouldn’t say that there is a “condition” called RCS. It is not surprising that patients may experience only partial control of symptoms, or progression of symptoms along with tumor progression. However, this is not really a distinct condition or syndrome. Also I would emphasize that measures such as increase in SSA dose/frequency or telotristat are primarily for patients with otherwise stable radiographic disease or very minimal tumor progression. I would also consider liver directed therapy to be appropriate for patients with stable, liver dominant disease but progression of symptoms. Other systemic treatments such as 177Lu-Dotatate, afinitor, etc. are more appropriate for patients with radiographic progression who may also experience symptom progression
5. Line 229: I’m not sure I would include pasireotide in this review since it is not approved for carcinoid syndrome and virtually never used for this purpose
6. Line 241: IFN alpha is an old drug but the term ancient (in English) implies that it originated during the Roman Empire (or before)…. It’s hard to say that IFN was optimally tested for tumor control: for example, we know now that OS is not a realistic endpoint, even with a large population
7. Line 272: “control of carcinoid syndrome with everolimus” is the title of an article. It is not a sentence. Furthermore, I would not consider case reports to be a ‘proper’ way to determine whether everolimus is an appropriate drug for carcinoid syndrome. Due to diarrhea being one of the side effects, it is probably not a great drug for this purpose.
8. Lines 360-375: I would say that radioembolization gained traction and then lost traction because the rate of chronic radioembolization-induced liver disease is almost certainly higher than what you quoted in reference 40. There is much more literature about this. Using radioembolization in unselected patient with bilobar liver metastases and good long-term survival prognosis is probably not a great idea.
9. Lines 443-453: There is literature from Pommier’s group in Oregon indicating that octreotide is not beneficial either for prophylaxis or treatment of carcinoid crisis. Please cite and comment. Wonn SM, Ratzlaff AN, Pommier SJ, McCully BH, Pommier RF. A prospective study of carcinoid crisis with no perioperative octreotide. Surgery. 2022 Jan;171(1):88-93. doi: 10.1016/j.surg.2021.03.063. Epub 2021 Jul 3. PMID: 34226047.
10. Line 593: The NANETS guidelines don’t establish a strict cutoff of urine 5HIAA for carcinoid syndrome screening. The precise language is …”we recommend that at a minimum all patients with significant elevations in serotonin or 5-HIAA levels (eg, >5× upper limit of normal) undergo annual echocardiography. Screening of patients with less prominent elevations of serotonin levels can be likewise considered.”
Author Response
Comments:
- Lines 157-165 and elsewhere: The frequency of carcinoid syndrome among patients with small bowel NETs is not well known. To start with, the diagnosis of carcinoid syndrome is often uncertain: e.g. a patient with diarrhea (which could be related to multiple factors e.g. bile salt malabsorption) and mildly elevated 5-HIAA. However, it seems to me that the analysis by Halperin et al, while often quoted, is a particularly poor method for ascertaining the frequency of carcinoid syndrome. This study was based on Medicare billing which is often random (e.g. a doctor may see a NET patient and automatically code for carcinoid syndrome, or see a patient with elevated CgA without a NET diagnosis and code for carcinoid syndrome). I would either delete this reference or point out its limitations. There are other institutional studies evaluating carcinoid syndrome frequency. For example: J Clin Oncol. 2013 Feb 1;31(4):420-5. doi: 10.1200/JCO.2012.44.5924, reported that approximately 70% of small bowel NETs (mostly metastatic) developed carcinoid syndrome over the course of their disease.
Response:
We agree with the Reviewer: the text has been modified accordingly. The suggested reference has been added to the references list (#18) (lines 163-173).
- Line 150: Ga68 dotatoc is not the only SSTR PET imaging modality. There is also Ga68 dotatate and Cu64 dotatate
Response:
Ga68 dotatate and Cu64 dotatate have been added (lines 151-152)
- Line 154 please rewrite this sentence: it is highly convoluted
Response:
The sentence has been rewritten (lines 156-159)
- Line 198: I wouldn’t say that there is a “condition” called RCS. It is not surprising that patients may experience only partial control of symptoms, or progression of symptoms along with tumor progression. However, this is not really a distinct condition or syndrome. Also I would emphasize that measures such as increase in SSA dose/frequency or telotristat are primarily for patients with otherwise stable radiographic disease or very minimal tumor progression. I would also consider liver directed therapy to be appropriate for patients with stable, liver dominant disease but progression of symptoms. Other systemic treatments such as 177Lu-Dotatate, afinitor, etc. are more appropriate for patients with radiographic progression who may also experience symptom progression
Response:
We thank the Reviewer for this valuable comment. We fully agree with this. The term “condition” has been removed. A comment has been added to the text (lines 244-254)
- Line 229: I’m not sure I would include pasireotide in this review since it is not approved for carcinoid syndrome and virtually never used for this purpose
Response:
The paragraph on Pasireotide has been removed, and Pasireotide is no longer mentioned in the visual abstract and Figure 1. A brief mention of Pasireotide has been mentioned in the text, highlighting that it is not approved for treating carcinoid syndrome (lines 251-254).
- Line 241: IFN alpha is an old drug but the term ancient (in English) implies that it originated during the Roman Empire (or before)…. It’s hard to say that IFN was optimally tested for tumor control: for example, we know now that OS is not a realistic endpoint, even with a large population
Response:
We apologize for the unclear meaning of this sentence. The term “ancient” has been replaced by “old,” and the sentence has been revised according to the Reviewer's suggestion (lines 268-273).
- Line 272: “control of carcinoid syndrome with everolimus” is the title of an article. It is not a sentence. Furthermore, I would not consider case reports to be a ‘proper’ way to determine whether everolimus is an appropriate drug for carcinoid syndrome. Due to diarrhea being one of the side effects, it is probably not a great drug for this purpose.
Response:
Again, we apologize for that. The sentence has been deleted (line 304)
- Lines 360-375: I would say that radioembolization gained traction and then lost traction because the rate of chronic radioembolization-induced liver disease is almost certainly higher than what you quoted in reference 40. There is much more literature about this. Using radioembolization in unselected patient with bilobar liver metastases and good long-term survival prognosis is probably not a great idea.
Response:
The paragraph on radioembolization has been rewritten (lines 400-409). References 43 (DOI: 10.1186/s12885-022-09302-z) and 44 (DOI: 10.2967/jnumed.121.263041) have been added
- Lines 443-453: There is literature from Pommier’s group in Oregon indicating that octreotide is not beneficial either for prophylaxis or treatment of carcinoid crisis. Please cite and comment. Wonn SM, Ratzlaff AN, Pommier SJ, McCully BH, Pommier RF. A prospective study of carcinoid crisis with no perioperative octreotide. Surgery. 2022 Jan;171(1):88-93. doi: 10.1016/j.surg.2021.03.063. Epub 2021 Jul 3. PMID: 34226047.
Response:
According to the Reviewer’s suggestion, a comment on the suggested reference has been added (lines 495-508). Ref #53 added (doi: 10.1016/j.surg.2021.03.063)
- Line 593: The NANETS guidelines don’t establish a strict cutoff of urine 5HIAA for carcinoid syndrome screening. The precise language is …”we recommend that at a minimum all patients with significant elevations in serotonin or 5-HIAA levels (eg, >5× upper limit of normal) undergo annual echocardiography. Screening of patients with less prominent elevations of serotonin levels can be likewise considered.”
Response: the text has been modified following the Reviewer’s comment (lines 650-653)
Reviewer 3 Report
Comments and Suggestions for Authors
The authors carry out a review of carcinoid syndrome in neuroendocrine tumors.
I consider it to be a good review. I did not detect any misplaced references or any missing references in the list of references. I only noticed a few lines that were left in bold.
Author Response
We thank the Reviewer for the positive reply to our manuscript. The text has been checked for text style and looks fine in the revised version.
Reviewer 4 Report
Comments and Suggestions for Authors
This is a comprehensive review on carcinoid syndrome covering all clinical aspects as well as treatments. It does not give much new information since there is not much recent news in the field.
Author Response
We thank the Reviewer for the positive feedback. We agree there is little novel literature to include on this topic.